# Psychosocial Risks in Teachers from Portugal and England on the Way to Society 5.0

**DOI:** 10.3390/ijerph20146347

**Published:** 2023-07-12

**Authors:** Ana Pimenta, Delfina Ramos, Gilberto Santos, Matilde A. Rodrigues, Manuel Doiro

**Affiliations:** 1Department of Industrial and Product Design, School of Design, Polytechnic Institute Cavado Ave, 4750-810 Barcelos, Portugalgsantos@ipca.pt (G.S.); 2School of Engineering, Polytechnic of Porto, ISEP, 4249-015 Porto, Portugal; 3Associate Laboratory for Energy, Transports and Aerospace (LAETA-INEGI), Rua Dr. Roberto Frias 400, 4200-465 Porto, Portugal; 4Algoritmi Research Centre/LASI, School of Engineering, University of Minho, 4800-058 Guimaraes, Portugal; mar@ess.ipp.pt; 5Center for Translational Health and Medical Biotechnology Research, School of Health of Polytechnic of Porto, 4200-072 Porto, Portugal; 6Department of Business Organization and Marketing, Vigo University, 36310 Vigo, Spain; mdoiro@uvigo.es

**Keywords:** COPSOQ II, education, psychological risks, occupational health, Society 5.0

## Abstract

Being a teacher is one of the most demanding jobs, as a result of this responsibility, these workers face many psychosocial risks. This study aims to characterize and compare psychosocial factors in Portuguese and British teachers and discuss how new developments in technology, namely digital technology can improve education and, in particular, contribute to fewer issues related to mental health. The Copenhagen Psychosocial Questionnaire Medium Version (COPSOQ II) was applied to the teachers of six Portuguese schools (three public schools and three private schools), three British public schools and three private schools with an international British curriculum (Switzerland, Spain and Portugal). The results showed that cognitive, emotional, and quantitative demands, as well as work rhythm and work/family conflict, are the key psychosocial factors among these teachers. Differences were found between the teachers of both countries. Some models are proposed, through the proposals of Society 5.0, for their minimization and/or removal. Society 5.0 is the vision of a new human-centered society in the fifth stage launched by Japan in April 2016, and it is cited in our study with the hope that it will contribute to solving many problems of today’s society.

## 1. Introduction

Mental health problems are one of the most relevant disorders in occupational settings [1,2]. It is estimated that mental health problems are responsible for around 50% to 60% of all lost workdays [1]. In fact, many psychosocial risk factors often have a high impact on people’s lives and lead to illnesses such as burnout, stress, anxiety, and depression [1,3]. One of the jobs in which there has been a significant increase in these problems is the teaching profession [2]. The mental health condition of teachers with different academic degrees continues to be a concern in different countries, as it can lead them to abandon the profession, as well as negatively influence students during the teaching-learning process [4]. Different risk factors have been contributing to this scenario, making it important to characterize them and implement appropriate measures to minimize psychosocial risks in the teaching profession.

It is undeniable that the world is changing rapidly due to the advancement of new technologies beyond the advancement of information technologies, leading us from the evolution of our society to the super-intelligent society called Society 5.0 [5]. This is a human-centered society that balances economic advancement with resolving social problems, and a system that integrates physical spaces and cyberspace [6,7]. Given this, the quality of life can be improved by applying Society 5.0 in education, especially for teachers in middle and secondary schools [6]. In Society 5.0, people expect social reform (innovation); with this, we will create a society that faces its future looking through its eyes, breaking the pattern that exists of being in stagnation. This will be a society where its members can show respect that will transcend generations, and all citizens can have an active and pleasant life [6].

In this study, we aim to characterize and compare psychosocial factors in Portuguese and British schoolteachers, identifying the main factors that affect teachers and providing possible solutions to minimize psychosocial risks according to the data evidence. These schools work mainly to ensure that students complete their secondary studies and do their assessments to join universities.

Through the comparison of the two countries’ realities and curriculums, we wish to discuss findings in light of Society 5.0, by understanding how new developments in technology can help education to become better, with fewer issues related to mental health. It is the opinion of the authors of this work that Society 5.0 will contribute to the creation of a better future, which is based on the most advanced technologies, but which will have the human being at its center. It promises the creation of new value through innovation and the simultaneous promotion of economic development and effective solutions to social challenges. It will free individuals from the countless constraints to which they are subject today and will focus on imagination and creativity. It will be an intelligent society capable of achieving the sustainable development goals towards the ideal society [5,6].

## 2. Literature Review

### 2.1. Psychosocial Risks in Education Teachers

Teachers are the key elements of educational services, as they are responsible for the development of educational programs and education of students at different levels. In their daily work, they deal with different demands. Their duties include planning and preparing classes, providing additional support to students, applying teaching-learning methodologies that ensure student’s participation and engagement, grading tests and documenting progress, assigning homework, providing individual support, attending several meetings with co-workers and students’ parents, promoting extracurricular projects, and completing other administrative tasks. Usually, a good teacher becomes an example for several students and a source of inspiration and motivation. However, it is essential to ensure their workability, particularly regarding mental health.

Psychosocial risks are of particular concern for teachers, since they have been related to high turnover and high absenteeism, which when referring to teachers have a negative impact on the students’ academic success [8,9]. A recent meta-analysis showed that burnout (exhaustion, depersonalization, and reduced accomplishment) and job satisfaction have an important role in teachers’ intentions to leave the profession [4].

According to the literature, teachers are exposed to psychosocial risk factors that can jeopardize their mental health. Psychosocial risks stem from aspects of how work is organized, social factors at work and in the work environment, equipment, and hazardous tasks. In this workgroup, having to deal with challenging students is a common concern, as identified in the report ESENER in 2019 [10]. In fact, some studies have shown the importance of students’ bad behavior on teachers’ mental health [11,12]. However, other psychosocial risk factors are also relevant; particularly those related to demands and stressors that teachers experience daily beyond students’ behaviors. Baeriswy et al. [11] emphasized conflicts with parents, workload, and prolonged working hours in emotional exhaustion in homeroom teachers. Time pressure and discipline problems were predictive of emotional exhaustion. Skaalvik and Skaalvik [13] verified that when teachers experience a feeling of value consonance, supervisory support, and positive relations with colleagues and parents, they report a feeling of belonging. Mijakoski et al. [14], through a literature review, identified different determinants of teacher exhaustion, including job satisfaction, work climate or pressure, teacher self-efficacy, neuroticism, perceived collective exhaustion, and classroom disruption.

Burnout is a highly prevalent phenomenon among teachers. For example, Bermejo-Toro et al. [15] showed that between 10 and 20% of teachers could suffer from high burnout levels and between 20 and 40% from moderate levels. Fernet et al. [16] stated burnout is predicted by changes in teachers’ perceptions of the school environment, in particular demands (e.g., work overload, role conflict) and resources (e.g., teacher efficacy, support by colleagues), and motivational factors (autonomous motivation and self-efficacy).

### 2.2. Society 5.0 and Its Role in Teachers’ Mental Health and Well-Being Improvement

Society 5.0 is the vision of a new human-centered society in the fifth stage launched by Japan in April 2016. It is cited in our study with the hope that it will contribute to solving many problems of today’s society.

Society 5.0 is technology-based, focused on people, and encompasses a multitude of ‘smart’ applications. The important thing here is that this model puts people right at the heart of it. Originally coined by the Japanese government, Society 5.0 stands for an intelligent, fully networked, and sustainable society [17]. In Society 5.0, people expect a better society than the current one, that is, a society that faces its future where its members will be able to show intergenerational respect, and citizens will be able to live better [6]. To be able to reach this point, it will be necessary to start deep reforms in several areas, namely, enterprise reform, individual reforms, and social problem solving.

The purpose of this system will be to use information and communication technologies to help in the process of improving everyone’s lives. As long as retirement ages are not extended without compromising the state of the economy, which stops when that happens, citizens have a good level of health, allowing them to enjoy their last years without constraints, for example, such as musculoskeletal injuries [18].

It is also the objective of Society 5.0 that all citizens have a quality of life. So, that their profession does not lead them to have health problems, whether physical, psychological, or mental during their career.

According to Salgues [5], keywords for Society 5.0 include: (i) adaptability, (ii) agility, (iii) mobility, and (iv) reactivity. These are linked to the fact that mutations, changes, and evolutions are constantly observed in our daily lives, reflected in our structure, knowledge, and skills. Adaptability, agility, and reactivity are crucial and require the implementation of the Industry 4.0 model, which implies the use of additive techniques that increase the consumption of a few resources for producing the goods. However, mobility has effects on means of transport and in our homes, and we see that we are increasingly in a more mobile and interconnected age. In this new society, Society 5.0 translates to a “new world”, in which exchange is the most important factor. There will be the concept of the primacy of issues involving economic exchanges and the primacy of ideas. In western democracies, what prevailed for years has been the transaction of goods, and its performance of the means of transport led to globalization, but the primacy of ideas is what will take precedence. For example, China in 1992 developed its soft power of primacy of ideas, and consequently today we see all of its economic development. France is also an example of this, in another field designated as cultural exceptions. This leads us to the concept that the export of ideas and knowledge has become a source of wealth in a period in which the export of goods is in decline and the demand for services is increasing [19]. In developed countries, namely in the United States of America, a high percentage of young law graduates encounter great difficulties in professional insertion. This happens with the help of AI (artificial intelligence). The internet of things and cybernetics, in some cases, offer legal advice in just seconds both for simple and more complex legal cases with an accuracy level of 90% (above 70% of human accuracy). As a result, it is expected that in the coming years, this profession will see a reduction in staff: 90% fewer lawyers and only those who are specialized will survive. It is expected, according to Salgues [5], that as early as 2030, the computer will be able to compete with human intelligence. It is easy to see, even nowadays, that our mobile phones already have image recognition software, which can be used for many functions: for example, looking for people, which is dethroning professional physiognomists.

Social acceptance, however, is a prerequisite to creating Society 5.0. Many fear change—or are skeptical about technological progress—and this must be addressed. There’s a growing fear that AI -powered automation will lead to mass redundancies, and some commentators have said that up to 800 million jobs will go by 2030. This can be a terrible reality, which we are approaching. Hence, the education system must be geared towards the new Society 5.0—both in terms of research and teaching. Universities and schools have a duty to adapt their educational programs for digital natives, particularly when it comes to preparing for tomorrow’s job market [17]. The implementation of appropriate measures for a new way of teaching, in which teachers are the main players, is an absolute necessity.

## 3. Materials and Methods

### 3.1. Study Design

This is a cross-sectional study, whose aim is the characterization of psychosocial factors among teachers from European public and private schools, using the Portuguese and British curriculum. The research questions that guided the study were based on schools that work primarily to ensure that students finish their secondary studies and take their assessments to enter universities, as described in the introduction. The difference between public schools in England and private international schools’ curriculum is that there is more coursework to cover, which will lead teachers in private international schools to have more pressure while teaching the British syllabus.

There are similarities between these schools’ countries. Public school teachers only do their contact time with their pupils and have free periods to cope with all work required from the job. Apart from their contact time with students, private school teachers have to cover lessons to replace their peers, exam invigilation, and a more extended shift day than public school teachers. Consequently, they have an increased workload after they leave school. The research questions that guided the study were: What are the main psychosocial risk factors that affect all teachers?; What are the main differences between the psychosocial factors among Portuguese teachers and British teachers?; What are the main differences between the psychosocial factors among private and public schools? The schools that participated in the study were chosen firstly because they met the criteria established for the study and also because the Heads were interested to know how their teachers were feeling at that time and wanted to assess the psychological state of their teachers due to an increase in sick leaves.

### 3.2. Instruments

The Portuguese middle version of COPSOQ II [20] was used to assess the psychosocial factors among all the teachers using the Portuguese curriculum. The questionnaire was applied in international schools. Teachers could be of different nationalities and they had knowledge of the British curriculum.

Psychosocial risk analysis was performed using questionnaires on the Google Forms platform to ensure data confidentiality. In response to the questionnaires, respondents responded privately. These answers were not disclosed to the hierarchical superiors of the educational establishment.

National and international public and private schools participated in the study, namely:3 Portuguese public schools with a Portuguese curriculum (North Region, Lisbon, and South Region);3 private schools with Portuguese curriculum (North Region, Lisbon, and South Region);3 public schools in the City of London with a British curriculum;3 private schools with an international British curriculum (Switzerland, Spain, and Portugal).

The selected schools are essentially dedicated to preparing students for the completion of secondary education and conducting exams for access to university education. These schools began their activities in different decades, but they all have the objective of promoting the academic success of their students in common. Emphasis was given to the number of students admitted to universities. It was decided that only teachers from the 3rd cycle and secondary education (teachers with students aged between 13 and 18 years old) would be studied at the school. This choice was based on the fact the tasks performed by this group of workers are exposed to greater psychosocial risks as they deal with a group of students with an average of 25 students during school periods.

The questionnaire was applied online, following a previous contact given the informed consent by the teachers. *The Copenhagen Psychosocial Questionnaire II* (COPSOQ II) was sent via email to be completed, which could be done on computers on the school premises or at home. These results were stored in the Google forms database.

The items of COPSOQ II are measured using a five-point Likert scale, and the results of the scales can be presented from one to five points or transformed using the cutting points of 2.33 and 3.66 in order to obtain a traffic light graphic [20]. 

The variables measured on a Likert scale were analyzed through the presented categories, while the quantitative variables were analyzed from the measured values, such as the average obtained for each question (for questions on a scale of 1 to 5, a value greater than 3 is greater than the midpoint of the scale), the standard deviation associated with each question representing the absolute dispersion of responses, the coefficient of variation illustrating the relative dispersion of responses, the minimum and maximum values observed for the answers given to the various questions. An internal consistency analysis was also carried out, allowing the study of the properties of measurement scales and the questions that integrate them. Cronbach’s Alpha is the most used model in the social sciences for checking scales’ internal consistency and validity, measuring how a set of variables represent a given dimension. An internal consistency coefficient value measured by Cronbach’s Alpha greater than 0.80 is considered adequate, and an internal consistency coefficient between 0.60 and 0.80 is considered acceptable. Statistical tests used in this study serve to ascertain whether the differences observed in the sample are statistically significant, that is, whether the conclusions of the sample can be inferred for the population. The value of 5% is a reference value used in Social Sciences to test hypotheses; it means that we establish the inference with an error probability of less than 5%. As the sample size is in these conditions, it will not be necessary to verify the assumption and parametric tests can be applied. As the groups under study can be considered significant, the parametric Student’s *t*-test is used to analyze a quantitative variable in both classes of a dichotomous qualitative variable to verify the significance of the differences between the means observed for both the groups of the dichotomous variable. The t-test poses the following hypotheses:◦**H1.** *There is no difference in means between the groups of the dichotomous variable.*◦**H2.** *There is a difference in means between the groups of the dichotomous variable*.

When the test value of the t-test is greater than 5%, the null hypothesis is accepted. That is, there are no differences between the two groups. The null hypothesis is rejected when the test value is less than 5%. Therefore, there are differences between the two groups. The use of the chi-square test is addressed; in the face of two nominal variables or a nominal and an ordinal variable, the appropriate test to verify the relationship between each pair of variables is the chi-square, in which we have the hypotheses:§**H3.** *The two variables are independent, that is, there is no relationship between the categories of one variable and the categories of the other*.§**H4.** *The two variables present a relationship between themselves, that is, there is a relationship between the categories of one variable and the categories of the other.*

The null hypothesis is rejected when the test value is less than 5% (0.05), concluding that the two variables are related. When the test value is greater than the 5% reference value, we cannot reject the null hypothesis that the two variables are independent; it is concluded that they are unrelated.

### 3.3. Participants

The sample encompassed 340 teachers from public (N = 170) and private schools (N = 170), following the Portuguese (N = 170) and British (N = 170) curriculum (Table 1). These teachers belong to European schools located in different countries: six Portuguese schools following the Portuguese curriculum, three of them private and the remaining three public; three British public schools located in London, following the British curriculum; and three international private schools following the British curriculum, and located in Switzerland, Spain, and Portugal.

We also collected the data about the age group of each teach for each specific school (Table 2). The schools chosen for the project were chosen because the students had similar backgrounds, high expectations, and engagement to work at school. So, if students do not differ so much, the teachers will face similar pressures that will introduce the same psychosocial risks. Also, regarding the differences between British and Portuguese curricula, the international schools deliver both syllabi (British and the local curriculum), and the differences between the syllabus created some problems between colleagues (teachers) arguing about who works more or has the most demanding syllabus. It was this behavior between peers (that could lead to psychosocial risks) that led to the present study.

## 4. Results

The global assessment of the psychosocial factors among the sample of all teachers of this study showed the presence of several risk factors, mainly related to the dimension work demands, but also present in the dimensions of interpersonal relations and work-individual interface. Also concerning are the results obtained through the health and well-being scales.

According to the respondents, the main risk factors, showed in red label in Figure 1 (from right to left), are those presenting the higher percentages of respondents, which were: 89.7% for cognitive demands; 80.6% for emotional demands; 76.5% for pace of work; 72.9% for quantitative demands; 70.6% for work/family conflicts; and 65% for labor conflicts. Among the health and well-being scales, exhaustion (or burnout), showed the higher percentage of unfavorable results (68.8%), which means a big risk.

There are issues that help to minimize the risk, of which we highlight: meaning of work, development possibilities, transparency of the role-played at work, and social community at work, among others.

As this study was carried out for Portuguese teachers and foreign teachers, we wanted to know if the health risks for Portuguese teachers were different from those of international teachers. Also, it is important to note that Portuguese schools (public or private) have a massive bureaucratic workload that puts more pressure on teachers’ shoulders, and they believe that this factor makes their life difficult.

For this analysis, we used descriptive statistics and T-tests to check if they are the same or if the risks are different. 

The horizontal axis of Figure 2 represents the means of all responses obtained for each case. An asterisk* has been added to the most significant results. Thus, by analyzing Figure 2, it is possible to confirm that significant statistical differences exist between these groups on some study scales while others are not significant. Therefore, for these results and looking only to those that exist, on statistical difference, we can state that for teachers in Portugal, when we compare with teachers from other countries, the main risks are, among others, emotional demands, cognitive demands, work/family conflict, labor conflicts, and exhaustion, that is, burnout. In comparison, the risks that involve more teachers in the international system are, among others: development possibilities, meaning of work, pace of work, social community at work.

It is possible to confirm that work conditions are varied, which can create tensions and different perspectives for teachers.

As this study was carried out for public and private schools, we wanted to know if the risks of public education teachers are different from those private education teachers.

It is necessary to be aware that when we speak about public schools, these are schools that have public funding from the local authorities. It’s called public because the taxpayer pays for it. When referencing private schools, which are paid for by the students’ parents or careers, the school provides the British curriculum if it is called private international. The factors: pace of work, quantitative demands, labor conflicts, horizontal trust, work/labor conflicts and exhaustion (or burnout), are more prevalent for teachers in private schools. The horizontal axis of Figure 3 represents the means of all responses obtained for each case. An asterisk * has been added to the most significant results. Thus, as it is shown in Figure 3, the differences observed are not statistically significant. In the sample, according to Figure 3, the factors cognitive demands, emotional demands, development possibilities, social community at work, quality of leadership, general health, sleeping problems, stress, and depressive symptoms are more prevalent for public school teachers.

The factor labor insecurity is more prevalent for teachers in private education. The factor “meaning of work” does not differ between the two groups.

Not all the differences observed are statistically significant. The ones with the * sign are the statistically significant. When we analyze the results given by Portuguese teachers, we can verify the risks do not differ much among all samples. When we analyze the differences between private and international public education, we find that from the statistical analysis, we can say that the proof value is less than 5% for Work/family conflict and Predictability.

From the obtained results, it is possible to verify that teachers are exposed to the same risks, but they can change in order. Still, they are the same, and the first seven are in decreasing order cognitive demands, emotional demands, pace at work, quantitative demands, work/family conflict, exhaustion (or burnout), and labor conflicts. However, countries have different bureaucracies and syllabuses in their education system, so the risks will be different.

When checking the risks regarding public or private management at school, it is possible to see that exhaustion (or burnout), and work/family conflict are present in all of them and are important factors. However, the Portuguese teachers have other associated risks, and they are different due to the organization and requirements of the school. Teachers that work with the British syllabus do not have statistically significant differences; only the general health and rewards level are very different.

## 5. Discussion

Many authors, from the most varied countries, have published studies on exhaustion work of teachers, which is teacher burnout. For instance, Li et al. [21] conducted a study in China designed to examine how achievement goals, burnout, and school context relate to beginning teachers’ turnover intention in China. The results showed that mastery goals and performance-avoidance goals were related to teachers’ turnover intention because of burnout. They made a study to examine how achievement goals, burnout, and the school context relate to teachers beginning to work to turnover their intentions due to reasons that lead them to burnout, such as mastery goals, performance-avoidance goals, time, pressure and discipline problems. 

Saloviita and Pakarinen [22] conducted a study on Finnish teachers. This study documented several associations between teacher burnout and background variables. This work is in line with the work presented here as it confirms the existence of burnout in teachers.

Prasojo et al. [23] comducted research on Indonesian education. The dataset presents a relationship for possible predictors of burnout. The study developed by these researchers suggests that emotional exhaustion (characterized by emotional and quantitative demands in this study), depersonalisation (in labour conflicts), and reduced personal accomplishment are the three components that lead people to develop burnout.

Hassan and Ibourk [24] conducted a study to test burnout and job satisfaction among Moroccan elementary school teachers. The results of the study confirmed the two-dimensionality of the burnout measurement scale.

Brown and Biddle [25] did a study in the USA in which the participants were teachers. The results showed that the professional protection factor of working in a positive school climate showed a negative indirect effect specifically on burnout.

From all the studies mentioned here, which also includes our study, we can demonstrate that teacher burnout is a serious problem at an international level. We know that the risks really exist, and we have to learn to minimize them [26,27].

Digital transition can have an important role in teachers’ mental health by reducing their workload. If the work becomes more accessible and more efficient, prolonging working hours, a relevant risk factor of emotional exhaustion, is avoided, and the teachers have more time to develop more timely health-promoting actions [11]. Despite recent changes in schools using online platforms, it is not expected to look to AI as a digital transition; however, it must be seen as a tool for teachers to decrease their workload. With the evolution towards Society 5.0, teachers must be able to get a better quality of life. It’s also a fact that using artificial intelligence, big data, and the internet of things does not only have benefits, and we are at the start of this transition, so this means that the systems are not as accurate as they should be and will also process tasks in a stringent line. So, although everything requires some teacher supervision, this will also lead to more workload depending on the subjects. Also, because there will be development of the system and leaving the traditional methods used in schools, this can lead to a more complex overall system that some teachers will not be able to follow and will leave some teachers behind (creating more psychosocial risks).

By the results obtained during our study, and other studies referenced, their mental health is affecting their quality of life. It is not difficult to explain that psychosocial risks like the pace of work, quantitative demands required from leadership, and work/family conflicts generated by the workload and labor conflicts with students and colleagues will significantly impact teachers’ mental health. This affects peace of mind and, in some cases, the interaction with pupils, colleagues and family can be quite demanding and challenging to manage. We cannot forget that people of all ages use sarcasm and irony, so they want to undermine others in a way that their language and actions cannot target them.

These risks are a relay that brings distress to their lives, but no one may notice it. However, they can be noticed by work colleagues, leadership, and also family members. All can see changes in their attitudes, such as error-making decisions and feelings of guilt and anxiety. Also, their physiological health shows signs like tiredness and different problems in their musculoskeletal system, which can lead to mistakes or accidents.

With the help of artificial intelligence and internet of things, perhaps it will be possible to reduce in a significant way the risks regarding quantitative demands and the pace of work that will lead to a decrease in work/family conflict. Therefore, Big Data allied to artificial intelligence can reduce the amount of work by providing regular updates to teachers. It will also help by having access to worksheets and assessments, assisting the teachers in knowing what each student’s weaknesses are to provide accurate feedback or resources that will help him to thrive. Of course, there is a need for an online platform that needs to be updated continuously and should have different languages.

Allowing teachers worldwide to be able to connect and work together will create support from colleagues even if at that school they do not have that support. It is also possible to develop new platforms with a lot more resources that can help teachers reach the goals required by their managers. One of the biggest causes of stress for teachers is marking assessments, especially when they have to mark exams that are important to provide access to universities. Due to different perspectives, different training, and because the teams that support the examiner markers sometimes do not provide an answer in useful time, this can produce unfairness when grading. Therefore, we can request cybernetics help to perform the correction of assessments. With this, we will be ensuring uniformity for all students using the same rules for everyone, and of course accelerating the process to make sure that results and feedback will be provided to students a lot faster. If we can decrease the workload for a teacher, they will able to do better work, providing a better balance in life with work and family. This, in turn, will lead to a decrease in stress eliminating depression symptoms, insomnia and exhaustion, or burnout. These work improvements will help them value their job, and not affect them in the way as it is does today. A double effect will be created: an increase in satisfaction for the teacher who will be a more effective teacher and will make sure that students will have better preparation for their future. On the other hand, it will be possible to decrease the amount of money spent every year in health care for these professionals minimizing the financial impact for governments.

## 6. Limitations and Future Research

As limitations of this work, we can consider the low number of schools and countries involved. As future research, we propose expanding the number of schools and countries, because as we see in this work, the psychosocial risks of teachers is a common problem in many countries.

## 7. Conclusions

It is possible to conclude that the risks teachers face are the same in any education system. Furthermore, those risks are cognitive demands, emotional demands, the pace of work, quantitative demands, work/family conflicts, exhaustion (or burnout), and labor conflicts. However, regarding countries/syllabus, those risks are for certain, different. Thus, we can conclude that the working conditions that cause these feelings of discontent or tension are different in different countries. Therefore, in Portugal, differences between the public and private system of education are statistically different, impacting teachers’ lives.

There are, however, some limitations to this study; several teachers from the private schools were not comfortable answering this questionnaire. Another aspect that influenced the results is the massive difference in how the teachers in Portugal get their jobs because they might have to distance themselves from relatives when moving country, increasing the risk of psychosocial risks. It was also necessary to complement this study with other analysis, like TISES (*Teacher Interpersonal Self-efficacy Scale*) and MBI-ES (*Maslach Burnout Inventory*™), to show more critical factors and their real impact.

It is necessary to help teachers to resolve problems like burnout and work/family conflict; with these problems dealt with, they will be able to develop a better life, which will lead to fewer emotional, psychological, and physical issues. Measures need to be taken to improve the life of these professionals. In conclusion, Society 5.0, with the help of robotics, Big Data, and artificial intelligence can be the chance to help teachers recover their mental wellness. However, it is imperative to start to think of them as vital to reduce the psychosocial risks to improve the quality of life for teachers.

## Figures and Tables

**Figure 1 ijerph-20-06347-f001:**
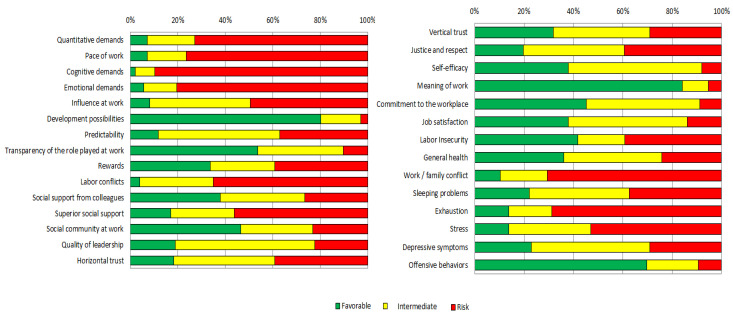
Description of COPSOQ II scales by a traffic light graphic among all teachers.

**Figure 2 ijerph-20-06347-f002:**
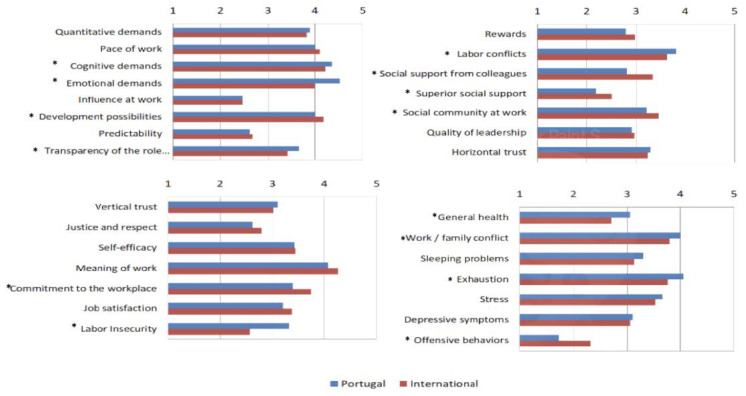
Statistical differences between groups in different countries. *:The ones with the * sign are the statistically significant.

**Figure 3 ijerph-20-06347-f003:**
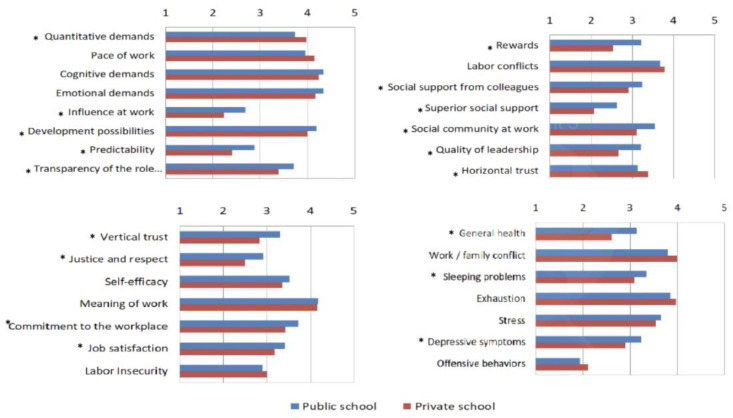
Statistical differences between public and private schools. *: The ones with the * sign are the statistically significant.

**Table 1 ijerph-20-06347-t001:** Sample characterization.

	Portuguese Curriculum		British Curriculum	
	Public School	Private School	Public School	Private School
Number of teachers	85	85	85	85
Gender	F M	F M	F M	F M
	74 11	74 11	80 5	84 1
Total	170		170	
	340			

**Table 2 ijerph-20-06347-t002:** Sample characterization in terms of age.

		Age			
		<30 Years	30 < Years < 40	40 < Years < 50	>50 Years
Portugal—Public	N	0	29	39	17
	% no Grupo	0.0%	34.1%	45.9%	20.0%
Portugal Private	N	10	44	25	6
	% no Grupo	11.8%	51.8%	29.4%	7.1%
International Public	N	24	33	21	7
	% no Grupo	28.2%	38.8%	24.7%	8.2%
International Private	N	22	32	14	17
	% no Grupo	25.9%	37.6%	16.5%	20.0%

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
