# Peer review of "Psychosocial Risks in Teachers from Portugal and England on the Way to Society 5.0"

_ijerph, 2023, doi:10.3390/ijerph20146347_

Round 1
Reviewer 1 Report
Thank you for giving me the possibility to review the paper " Psychosocial risks in teachers from Portugal and England and the role of 5.0 society". The present study deals with an interesting topic. The paper is within the topic of the journal. However, a large portion of the manuscript need to be improved and I have made suggestions.
Introduction:
1. Page 1, line 31; Page 2, line73
Are the authors willing to cite references such as the article in these texts? If so, please implement the appropriate citation method.
2. Page 1, line96-
Please describe in detail, with references to the literature, why applying Society 5.0 to Education will improve the quality of life.
3. Page 2, line54-
The authors have written a chapter on the literature review, but the descriptions here should be logically related to the descriptions in "1. Introduction". In particular, the authors need to clearly state how Society 5.0 will contribute to this study.
4. Page 2, line56-65
After listing the duties of teachers, the authors state, "However, it is essential to ensure their workability, particularly regarding mental health." The authors should add a sentence citing literature that shows that the mental health of teachers is harmed by those duties.
5. Page 2, line76
Is the correct phrase by the authors "in mental health disorders of teachers"?
6. Page 2, line80-92
The sentence structure does not seem logical. I suggest the following sentence structure. Please modify the order of references as appropriate.
>> Time pressure and discipline problems were predictive of emotional exhaustion. Skaalvik and Skaalvik [14] verified that when teachers experience a feeling of value consonance, supervisory support, and positive relations with colleagues and parents, they report a feeling of belonging. Mijakoski et al. [15] through a literature review, identified different determinants of teacher exhaustion, including job satisfaction, work climate or pressure, teacher self-efficacy, neuroticism, perceived collective exhaustion, and classroom disruption.
Burnout is a highly prevalent phenomenon among teachers. For example, Bermejo-Toro et al. [16] showed that between 10 and 20% of teachers could suffer from high burnout levels and between 20 and 40% from moderate levels. Fernet et al. [13] stated burnout is predicted by changes in teachers’ perceptions of the school environment, in particular demands (e.g. work overload, role conflict) and resources (e.g. teacher efficacy, support by colleagues), and motivational factors (autonomous motivation and self-efficacy).
7. Page 2, line93-
“2.2.” (regarding Society 5.0) is not directly related to the data in this study. It is recommended that it be stated briefly.
Materials and Methods:
8. Page 4, line156-160-
These sentences should be included in the Introduction. The authors should describe the reasons that led to this research question based on the description in the Introduction.
And, the authors state "What are the main differences between the psychosocial factors among Portuguese teachers and International teachers? I think it needs to be rephrased as British teachers, not International teachers, accurately.
9. Page 4, line170-178
The authors should describe the criteria by which they selected these schools and how they recruited them.
10. Page 4, line192-194
The authors should describe COPSOQ II in detail. For example, the classification of Favorable, Intermediate and Risk should be described in detail. Also, reference 20 seems to be a document written about COPSOQ, can the same cutoff values be used for COPSOQ II?
11. Page 5
The authors should provide a detailed description of the statistical analysis at the end of Materials and Methods.
12. Page 5, Table 1
The authors should list the basic demographics of the subjects, such as age and gender, in Table 1.
Results:
13. Page 5-7
The authors provide interpretation and discussion in the Results section. Authors should objectively describe their data in the Results section.
The relevant sections are as follows: Page 5, line205-209; Page 5, line216-218; Page 5, line222-224; Page 6, line235-236; Page 6, line239-240; Page 7, line257-268.
14. Page 6, line250-254
Which analysis do the results discussed in this paragraph reflect?
15. Page 6, Figure 2
Please add in the figure what the horizontal axis in Figure 2 means. Please modify the figure to show the results of the t-test by adding an asterisk (*) to the items for which statistically significant differences were found. If the data are not normally distributed, correct the test method.
Regarding the above description, the authors should also apply it to Figure 3.
Discussion:
16. Page 7, line 270-286
The authors should not just list the results of previous studies, but compare them to the results of this study. Then, please describe the position of this study in the context of all studies.
17. Page 8, line 298-324
The authors should provide a discussion based on the results of this study and citing the literature.
Conclusions:
18. Page 8, line 326-
The authors should briefly state the facts found and suggested, and the prospects for future research in the Conclusions section.
Author Response
The authors are grateful for all suggestions for improvement. Our response is attached.

Reviewer 2 Report
Line 32, 214, 231: unnecessary capital letter in burnout
Line 36-37: „it may lead them to leave the 36 profession and to negative impacts on the teaching-learning proces” – I would recommend to rewrite the sentence, as it is not clear why it may lead tchem (whom?) to negative impacts
Line 46: unnecessary capital letter in education
Line 90: spelling mistake in burnout
Line 98, 110, 120, and further where relevant : Inconsistent use of capital (here small) letter in Society 5.0
Line 132: should be „offer” if it refers to plural
Line 135: Are you sure of the pecentage? 910% fewer lawyers?
Line 136: according to the same author [19] and others – but who is „others” (no reference)?
Line 154 and 160: public & private schools are mentioned, whereas in abstract only public schools are mentioned
Line 159: International or British teachers? It is not clear what were the differences between both the teachers and the schools? Too many factors that are not clearly describes (see also comments to Line 164)
Line 164, 170-178: Were all schools international? There was no research question on the international schools, which might casuse a difference. It is not clear what is the difference between 3 public schools in the City of London with a British curriculum and 3 private schools with an international British curriculum, and what criteria were used to compare them with Portuguese public and private schools. What is the difference between a British curriculum, an international British curriculum, and Portuguese public and private schools’ curriculum?
Line 166: Google Forms/Google forms – consider that form
Participants section: Are these location comparable? SInce the location of the schools vary so much, there may be seveal factors influencing the psychosocial risks (among which public vs. private, Portuguese vs. British might not be the crucial ones). Therefore Portuguese curriculum vs. British curriculum comparison (opposition) is probably insufficient.
Line 206: related to
Line 230: other countries
Description of Figure 2. Statistical differences between groups in different countries. It i stoo general. What does it mean „different countries” in this context? I understand that they are:” three international private schools following the British curriculum, and located at Switzerland, Spain and Portugal” (lines 201-202). However, it should be better expalined why these countries were chosen, what are the possible differences between the school settings in these 3 countries. Why should public and private schools in Portugal differ so much? In line 243 you wrote: „As it is shown in figure 3, the differences observed are not statistically significant”. Maybe there are other psychosocial factors that could reveal differences.
Line 252: What does „international public education” mean? Does school with British curriculum located in Portugal belong to this category? What makes the school international: location or curriculum?
Line 256: Figure 3. Statistical differences between public and private schools. The same remarks as to the above Figure 2.
Line 264: exhaustion – unnecessary capital letter
Line 277: What king of background variables they studied? Please elaborate on this further. Sounds interesting.
Line 279: Prasojo et al. [23] did research on Indonesian education. The dataset presents a rela- 279 tionship for possible predictors of burnout. Please extend the discussion. It is too brief and too general.
Line 281-283: Hassan e Ibourk [24] fizeram um estudo para testar o esgotamento e a satisfação no trabalho entre professores marroquinos do ensino fundamental. Os resultados do estudo confirmaram a bidimensionalidade da escala de medição de Burnout.
Line 290: Digital transition can have an important role in teachers' mental health by reducing their workload. – The presented study did not refer to digita transition (it was not studied as such, only conclusions are drawn). I would suggest a change of the title, since the COPSOQ II scales is not about digital transition and (as far the presented results are concerned) teachers were not asked or monitored in the context of digital transition. We have no information how advanced each school was with digial transition.
Line 294-295: By the results obtained during our study, and other studies referred, their mental health is affecting their quality of life. In the study, we have no analysis regarding this issue (the impact of mental health on teachers’ quality of life).
Line 296-297: These risks are a relay that brings distress to their lives, but no one can notice it. Not clear who should notice what? Not clear.
Discussion section: Digital transition should be also discussed from the perceptive of its challenges and drawbacks, not only its benefits. The discussion is one sided. Must be improved. Moreover, is AI really equal to digital transition?
Line 326: education
Line 329-330 & 331-333:
However, regarding countries/syllabus, those risks are for sure, different. Therefore, in Portugal, differences between the public and private system of education are statistically different, impacting teacher’s lives.
How about insigificant differences that were reported in the study? Not clear. Do authors comment on their own research?
Line 344-346: In conclusion, Society 5.0, with the help of robotics, Big Data, Artificial intelligence can be the chance to help teachers recover their mental wellness. This sentence does not justify the title of the article, which is „Psychosocial risks in teachers from Portugal and England and 2 the role of 5.0 society”. The role of 5.0 society (or Society 5.0?) was not really studied. The title should be changed.
Author Response

(The authors gave the same response as above.)

Round 2
Reviewer 1 Report
Thank you for the opportunity to re-review this manuscript. The authors have largely revised the paper in accordance with the reviewers' comments. However, I believe that a few modifications are necessary to make the paper more readable for people in many research areas.
For Figures 2 and 3, I can also agree with the authors' opinion, “Just look at the bars and everyone sees the results”. However, it would be easier for the reader to understand the results if the figure shows which items have statistical differences between them. And the title of the figure states "statistical differences". So I suggested adding an asterisk in the last peer review. I am an epidemiologist, so the above is standard. I leave the decision on this matter to the authors and editors.
In addition, for Figures 2 and 3, if the horizontal axis means percentages, could you add (%) at the upside?

Author Response
Many thanks for suggestions

Reviewer 2 Report
I accept the present form of the paper.
It needs only small corrections when it comes to fonts.
Author Response
Thank you very much.
